# NerfBaselines: Consistent and Reproducible Evaluation of Novel View Synthesis Methods

**Jonas Kulhanek**
CTU in Prague and ETH Zurich
Czech Republic and Switzerland
`jonas.kulhanek@cvut.cz`

**Torsten Sattler**
CTU in Prague
Czech Republic
`torsten.sattler@cvut.cz`

**Implemented datasets**
- Standardized dataset format and coordinate systems
- Implements common loaders, e.g. COLMAP, NerfStudio, Bundler, etc.
- Standardized evaluation protocols matching the original papers

**Integrated methods**
- Unified interface for NeRFs and 3DGS
- Original codebases are used such that reported metrics match the papers
- Each method installed into isolated encapsulated environment

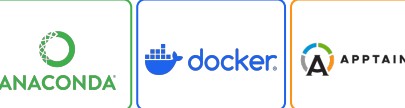

**NerfBaselines**
- Unified API to access methods
- Better reproducibility
- Simplified installation
- Online benchmark

**Isolated encapsulated environments**

ANACONDA    docker    APPTAINER

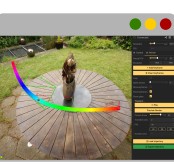

**Interactive viewer**

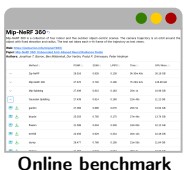

**Online benchmark**

## Abstract

Novel view synthesis is an important problem with many applications, including AR/VR, gaming, and simulations for robotics. With the recent rapid development of Neural Radiance Fields (NeRFs) and 3D Gaussian Splatting (3DGS) methods, it is becoming difficult to keep track of the current state of the art (SoTA) due to methods using different evaluation protocols, codebases being difficult to install and use, and methods not generalizing well to novel 3D scenes. Our experiments support this claim by showing that tiny differences in evaluation protocols of various methods can lead to inconsistent reported metrics. To address these issues, we propose a framework called NerfBaselines, which simplifies the installation of various methods, provides consistent benchmarking tools, and ensures reproducibility. We validate our implementation experimentally by reproducing numbers reported in the original papers. To further improve the accessibility, we release a web platform where commonly used methods are compared on standard benchmarks.
**Web:** `https://jkulhanek.com/nerfbaselines`

## 1 Introduction

Due to the explosive growth in the number of NeRF-based and 3DGS-based works, with about 10 new papers citing NeRF [37] and 3 papers citing 3DGS [19] each day, there isn't a single evaluation framework and the individual works are based on different codebases for evaluation, resulting in numbers that sometimes are not directly comparable. Different methods sometimes use different evaluation protocols for the same datasets and often use specific data processing or hyperparameters which do not translate well to novel datasets (not used in the original papers). Furthermore, methods are often difficult to install with conflicting dependencies, and the codebases differ significantly,

Submitted to the 38th Conference on Neural Information Processing Systems (NeurIPS 2024) Track on Datasets and Benchmarks. Do not distribute.

which makes custom visualization or benchmarking laborious. To address these issues, we propose a framework designed to standardize and simplify the evaluation and comparison of these methods.

Not all papers follow the same evaluation protocol (partly because it is not fully specified) and sometimes innocuous changes can have a significant impact on the overall results. Examples include methods using different image resolutions for evaluation, different parameters for image downscaling, or different parameters for computing metrics. In our experiments, we show that even small discrepancies can cause substantial differences in the metrics, validating the need for a framework such as NerfBaselines. Our framework standardizes evaluation protocols, following best practices for each benchmark dataset.

Even with access to original source codes, reproducing paper results can be challenging due to codebases evolving over time or some codebases becoming obsolete where it is no longer possible to install the dependencies. To tackle this, for each method, we provide a consistent and reproducible way of installation (by encapsulating the runtime environment) which enables us to release checkpoint that can be used to render from custom camera positions without the need to rerun the training. Furthermore, we release a website with benchmarks comparing various methods on multiple datasets.

Many existing codebases cannot be easily applied to novel datasets due to having simplifying assumptions in the code, e.g., not implementing various camera models, requiring uniform camera intrinsic, or image sizes being equal. Various methods also use different interfaces, dataset formats, and coordinate systems, making integration and custom rendering laborious. Therefore, in NerfBaselines, we have integrated some important and influential codebases and implemented the missing functionality. Thankfully, most codebases are derived from the few original repositories (often with minor modifications), and therefore, they can also be integrated easily. Our experiments show that our integrated methods match the original ones with sufficient precision.

Furthermore, most methods compare using a set of metrics (e.g., PSNR) computed on a set of images taken as a subset of the training trajectory, which does not necessarily correlate well with the performance outside of the training trajectory [29]. Rendering videos from custom trajectories is often much better at demonstrating multiview consistency. Therefore, we release a web-based camera trajectory editor, based on Viser [53], to design camera trajectories and standardize their format. The common interface allows all integrated methods to render novel images from these trajectories.

In summary, to simplify benchmarking and improve reproducibility, we propose:

- A unified interface for NeRF and 3DGS methods, standardizing dataset formats and evaluation protocols.
- The NerfBaselines framework, which installs each method in an isolated environment to manage dependencies and ensure reproducibility. We experimentally verify our integrated methods reproduce results from original papers.
- A web-based camera trajectory editor and tools to render different camera trajectories, offering a more comprehensive assessment of performance beyond traditional metrics like PSNR, SSIM [6], or LPIPS [70].
- A web platform for comparing the performance of various methods on different datasets, providing checkpoints, and enabling interactive viewing of results.

## 2 Literature review

In recent years, progress in novel view synthesis has enabled real-time photo-realistic rendering of images from novel camera viewpoints. There has been a surge of interest, first with the advent of neural radiance field methods (NeRFs) [37, 30] which enabled photo-realistic results and a wide range of application. The second wave of attention came with the introduction of 3D Gaussian Splatting [19] , matching NeRFs in terms of the rendering quality while enabling real-time rendering.

**Ray-based methods.** NeRFs represent a scene using a continuous volumetric radiance field, originally parameterized by a fully connected neural network [37, 2]. While this method enables high-quality

rendering for forward-facing or bounded and object-centric scenes, it initially suffered from aliasing issues, which were addressed in [2] and extended to unbounded scenes in [3, 69]. However, the MLPs used in NeRFs posed a bottleneck in terms of both training and rendering speeds. To address this, caching of the radiance field was proposed to enable faster rendering [16, 45], though this came at the cost of large storage requirements. Consequently, several methods have investigated replacing MLPs with alternatives such as *sparse grids* [14, 52], *point clouds* [61], *tetrahedral meshes* [24], *tensorial factorizations* [9, 15, 46], or *hash-grids* [39, 4, 56], often combined with tiny MLPs on top of the internal representation. While each approach has its advantages and disadvantages, hash-grids [39] have become the most popular representation due to their speed and scalability, with Zip-NeRF [4] achieving SoTA performance by addressing the aliasing issues of grid-based representations.

NeRFs have been extended in various ways, such as handling dynamic scenes [41, 44], modeling different image appearances [33, 15], and accommodating data with imprecise poses [5, 42]. They have also been applied to model semantics [21, 22] and for style transfer [54, 17]. To better model geometry, some methods replace the radiance field's density with a signed distance function (SDF) [55, 57, 63]. Finally, another line of work focuses on extending these methods to generative [11, 43] or sparse-view settings [66, 8, 26], where the method is trained on a class of environments and then performs novel view synthesis given a few context images.

**Rasterization methods**, which project 3D geometry onto 2D image planes [19, 13], can be fast thanks to hardware optimization (rasterization pipelines have been dominant in the past). Recently, Gaussian Splatting (3DGS) [19] has gained popularity by matching the rendering quality of NeRFs while rendering at real-time speeds. 3DGS represents scenes using 3D Gaussians, which are projected onto the image plane [72], rasterized, and alpha-composited to render a view.

Like NeRFs, 3DGS has been modified to handle aliasing issues [67, 31], extended to larger scenes [20], and optimized to reduce the size of the representation [40] and to improve its geometric accuracy [18]. A minor change to the adaptive density control (ACD) was proposed in [65, 68] – where the sum of gradients was replaced with a sum of its norm , and a faster initialization method was proposed in [12]. Additionally, 3DGS has been applied to various domains, including SLAM [34] and physics simulation [59]. It has been extended to handle dynamic scenes [32] and semantics [71] using an N-D rasterizer. Generative 3DGS has also been proposed [10].

Unlike ray-based approaches which sample pixels randomly from training views, 3DGS optimizes on full images, requiring different training batch formation. In NerfBaselines, we do not enforce any specific way of constructing training batches, allowing each method to use the format it needs.

**NerfStudio** [53]
- **Tetra-NeRF** [25]
- Instruct-NeRF2NeRF [17]
- LERF [21]
- Nerfbusters [58]
- Splatfacto [64]

**Multi-NeRF** [3, 2]
- **Zip-NeRF**
- NeRF [37]
- Mip-NeRF [2]
- RawNeRF [38]
- **MipNeRF-360** [3]
  - SeaThru-NeRF [28]
  - NeRF on-the-go [47]

**Gaussian Splatting (INRIA)** [19]
- **Mip-Splatting** [67]
  - **Gaussian Opacity Fields** [68]
- AbsGS [65]
- ScaffoldGS [31]

**NeRF** [37]

**Instant-NGP** [39]

**TensoRF** [9]

Plenoxels [14]

Figure 1: **Existing codebases.** Integrated methods are **bold green**.

## 2.1 Existing codebases

Most current methods are based on a few core repositories: NerfStudio [53], Multi-NeRF [2, 3], Instant-NGP [39] for NeRFs, and Gaussian Splatting (INRIA) [19] for 3DGS, typically with moderate modifications. Therefore, in NerfBaselines, we focused on integrating these core repositories, as it simplifies the subsequent integration of derivative works. In Figure 1, we illustrate the relationships between popular repositories and highlight those currently integrated with NerfBaselines.

**NerfStudio** is a popular framework that introduced the modular separation of NeRFs into components such as ray samplers, radiance field heads, etc. It supports various dataloaders, camera types, and export formats. Unfortunately, the rapid evolution of NerfStudio introduces frequent breaking changes. Therefore, in NerfBaselines, we freeze each method to ensure reproducibility.

**Multi-NeRF** is fully implemented in JAX and does not have custom CUDA kernels (unlike Instant-NGP or NerfStudio), making it easy to install, but slower. Early versions based on Mip-NeRF360 supported only a single camera per dataset and a single image size; therefore, in NerfBaselines, we have extended these methods to handle more complex datasets.

**Instant-NGP** is a highly optimized implementation that has inspired numerous follow-up works [4, 53, 21, 22]. However, it is a less popular choice as a codebase due to its C++ training code being more difficult to extend. Since the repository does not natively support rendering with custom (distorted) cameras, in NerfBaselines, we have implemented this functionality.

**Gaussian Splatting (INRIA)** is the most popular choice for 3DGS methods, primarily because (until recently) it was unmatched in performance. The repository only supports pinhole cameras with their centre of projection being in center of the image. We have extended it to work with arbitrary camera models, performing undistortion/distortion for more complicated camera models.

# 3 Framework design

When designing the NerfBaselines framework, we aimed to address common issues when benchmarking novel view synthesis methods. These include: **different interfaces**, where each codebase has a different structure, making it difficult to interface with it. To this end, we propose a unified interface that all methods can share. **Installation challenges** arise as various methods often require specific and potentially difficult-to-install dependencies. To simplify the process, we install each method in a separate environment where we fully control the dependencies. For **fair evaluation**, we implement a standardized evaluation protocol for all methods, closely matching the original protocols proposed with the datasets. **Reproducibility** is ensured by carefully storing and tracking checkpoints used to generate predictions and compute metrics. Additionally, we have implemented a viewer with a trajectory editor and a public website to compare existing methods on standard benchmark datasets.

**Unified method API.** The different codebases [53, 3, 19, 9, 19] have very different code structures making it difficult to interface with them in a unified way. While studying the codebases, we identified a common structure which can be shared between all existing methods, regardless if they are raycasting-based (as in the case of NeRF codebases [53, 3, 19, 9, 60]), or rasterization-based (as in 3DGS methods [19, 20, 67, 68, 64]). For each codebase, we define a class called `method`, which encapsulates all that is needed to train the method and render new images from an existing checkpoint. Each method implements (among others) methods: `train_iteration` - which performs one training step, and `render` - which renders images from input cameras. Additionally, if the method supports appearance conditioning [33, 53], it implements the functionality to obtain the embeddings from unseen images. The detailed interface is given in the *supp. mat.*

**Environment isolation.** Installing methods is often difficult due to a set of hard-to-install and sometimes obsolete dependencies, e.g., CUDA, CuDNN, GCC, or OpenGL, where the version must match. The dependencies are sometimes in conflict with each other or in conflict with the user's environment. Therefore, we provide a level of isolation, where each method is installed in its own isolated environment, allowing for different/conflicting dependencies without polluting the user's environment. The user then communicates with the isolated environments using interprocess communication. NerfBaselines then handles the installation of all necessary dependencies the first time a user uses a method. We provide three levels of isolation: Anaconda [1], Docker [35], and Apptainer [27]: **1)** While Anaconda [1] is commonly used in research and is easy to install even on HPC clusters, it does not offer a good level of isolation, as some dependencies are inherited from the system, e.g., glibc, OpenGL, and, therefore, may fail on some systems. **2)** Docker [35] provides the best level of isolation, fully encapsulating the user environment (except for the CUDA drivers needed to control GPUs). Unfortunately, it is more difficult to install and is not commonly present on HPC clusters. Therefore, we also support the alternative **3)** Apptainer [27] with a better level of isolation than Anaconda that is more commonly present on HPC clusters.

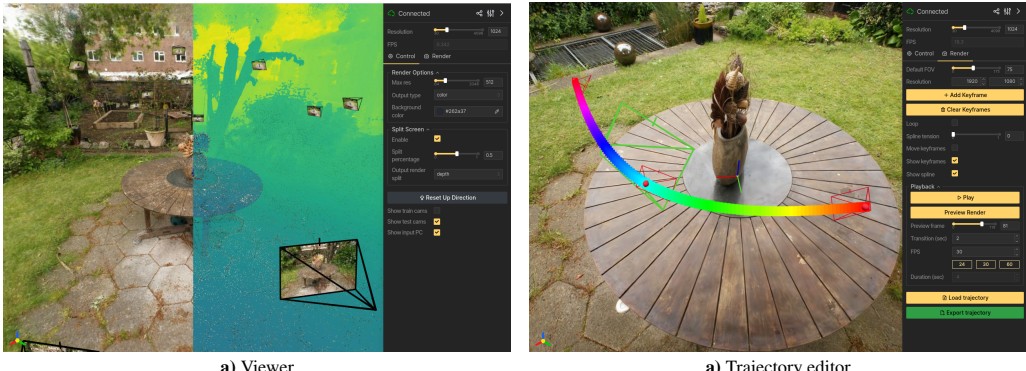

**a)** Viewer                                                   **a)** Trajectory editor

Figure 2: The **NerfBaselines Viewer** enables interactive rendering, shows train/test cameras, and input point cloud. It also has a trajectory editor (shown in the left figure).

**Unified dataloader.** Different codebases store and load data in various ways, complicating the process of integrating and evaluating methods across different datasets. For example, they use different downscaling algorithms and resolutions, different background colors, etc. The camera poses processing pipeline is also important, where methods like Instant-NGP [39] and NerfStudio [53] are sensitive to the scale of the scene, which depends on the camera placement relative to the scene's geometry. We propose a unified interface for dataset loading and processing to address these issues and to facilitate the transfer to new datasets. We implement support for commonly used dataset formats such as COLMAP [49, 48], NeRF's transforms.json [37], NerfStudio's transforms.json [53], Bundler [51, 50] (also supported by RealityCapture [7]), NeRF in the Wild Photo Tourism [33, 50] splits format, Tanks and Temples [23], and the LLFF dataset format [36]. The loaders ensure consistent data processing and provide a uniform interface and camera format for all datasets, thereby simplifying integration and evaluation across diverse datasets. It also makes the development of new methods easier as they do not need to implement their own dataloaders anymore. Furthermore, to simplify the ease of use, we support an automatic download of commonly used benchmark datasets such as the Blender dataset [37] and the Mip-NeRF 360 dataset [3].

**Viewer & camera trajectory rendering.** For novel view synthesis methods, evaluation is often conducted on a set of test images that are typically a subsequence of the training views. Unfortunately, this approach does not adequately demonstrate the method's robustness and multiview consistency in capturing the 3D scene [29]. Rendering a video from a custom trajectory from truly novel viewpoints farther away from the training images provides a more insightful evaluation. While the unified interface presented in Section 3 'Unified method API' significantly simplifies the process of rendering such trajectories, we further enhance this capability by providing an interactive viewer. This viewer allows users to inspect how the method performs outside the training camera distribution and includes a camera trajectory editor for designing custom trajectories. The viewer, based on NerfStudio's viewer [53] and utilizing the Viser platform [53] for the web interface, is depicted in Figure 2, where the viewer interface is shown on the left and the trajectory editor on the right. The trajectory editor also enables exporting the trajectory in a format suitable for subsequent rendering.

**Fair evaluation & Reproducibility** Evaluating and comparing different methods is challenging because original codebases often use different evaluation protocols. For example, some methods use different parameters for SSIM [6] and different LPIPS [70] architectures (like AlexNet vs. VGG). Some methods even use different image resolutions [60, 62], or compute metrics on the raw image (float range) as opposed to rounding to the `uint8` range – which cannot be reproduced from the commonly saved images. Therefore, we standardize the evaluation protocol by fixing the parameters of the metrics, resolution of images, and other aspects of the evaluation.

Sharing checkpoints often greatly improve research reproducibility, enabling validation of predicted test images and rendering 3D scenes from novel viewpoints. To enable this, NerfBaselines stores

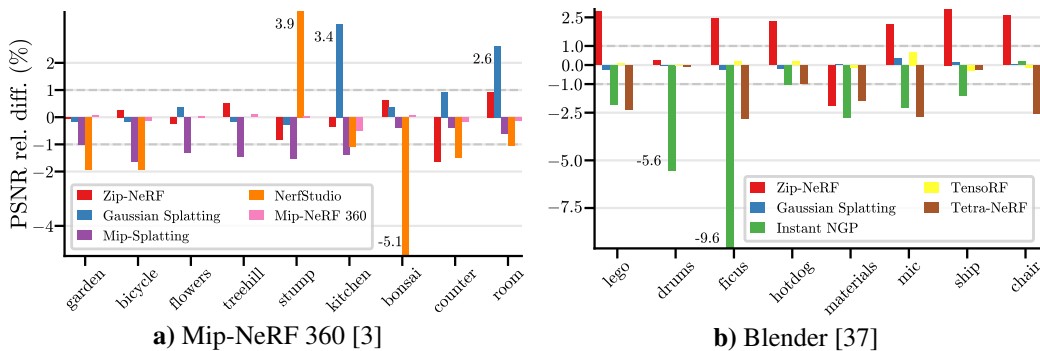

**a)** Mip-NeRF 360 [3]  **b)** Blender [37]

Figure 3: **Mip-NeRF 360 [3] and Blender [37] results** comparing PSNRs obtained via NerfBaselines with those reported in the original papers. We show the relative difference in %. In most cases, the difference is $< 1\%$. Instant-NGP [39] and Mip Splatting [67] are consistently underperforming because different evaluation protocols were used in the papers.

checkpoints to ensure models can be restored later. During method evaluation, we store the checkpoint SHA and verify it matches the released checkpoint. For all methods, we fix random seeds to match the official implementations. Additionally, for each integrated method, we provide a set of tests to confirm that loading the checkpoint genuinely reproduces the results.

**Web platform & Extensibility.** As part of the NerfBaselines toolbox, we release a website where methods are compared and their numbers (measured with NerfBaselines) reported. It also enables the checkpoints and the predictions to be downloaded for all scenes. The web can be found at `https://jkulhanek.com/nerfbaselines`.

Finally, NerfBaselines is designed in such that it makes it easy to extend it with new methods and datasets. Therefore we provide extension points to register new datasets, new evaluation protocols, and add new loggers (currently, we implement wandb and tensorboard loggers). The new method can be added by implementing the protocol described in Section 3 'Unified method API' and providing a specification file, containing the installation script and method's metadata.

## 4 Evaluation

In our experiments, we **1)** verify that the integrated methods match the original metrics, **2)** show the tradeoff between quality and training cost, **3)** motivate the need for standardized evaluation protocols by demonstrating that small changes in evaluation protocol can bias results, **4)** demonstrate transferability to novel datasets, and **5)** qualitatively compare methods outside training trajectories. All our experiments used NVIDIA A100 GPUs. A single GPU was used for all but Mip-NeRF 360 [3], which used four GPUs. For comparisons, we use PSNR, SSIM [6], and LPIPS [70] (AlexNet). In the main paper, we mostly report PSNR, while other results can be found in the *supp. mat.*

### 4.1 Reproducing published results

First, to validate our framework, we reevaluate important methods on the standard benchmark datasets: Mip-NeRF 360 [3] and Blender [37]. We use the same evaluation protocol for all methods. The results are compared to the original numbers as published in the papers in Figure 3, with detailed numbers given in the *supp. mat.* Note, that in the figures and tables, we only compare with methods that released their numbers on the datasets in the corresponding publications.

**Mip-NeRF 360 results.** As shown in Figure 3, NerfBaselines reproduces the original results with a deviation of less than 1% for most scenes. For Mip-Splatting [67] and 3DGS [19], the difference in the numbers was caused by the different evaluation protocols used as discussed in Section 4.3. In the case of NerfStudio [53] and Tetra-NeRF [25], the codebase evolved since the time of the release which is likely the cause of the difference.

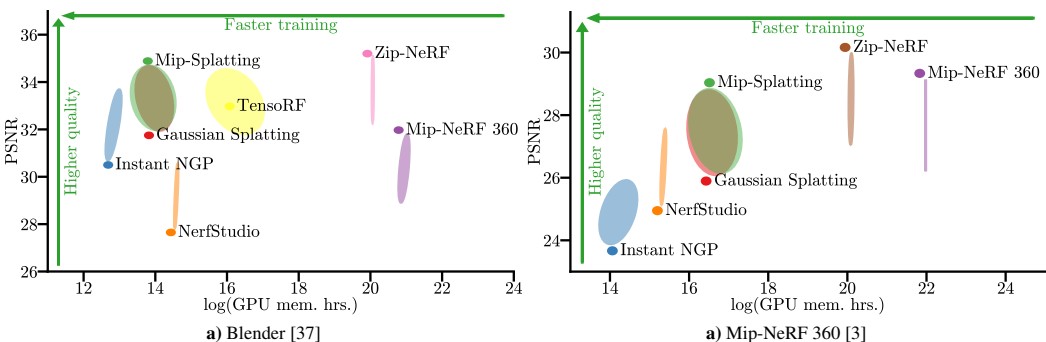

Figure 4: **Performance vs. training cost tradeoff.** We compare different methods' rendering quality (PSNR↑) and rendering cost (GPU mem. hrs.↓). The variance across different scenes is visualized by the ellipse scale. While Zip-NeRF [4] performs the best, it is more costly to train than 3DGS[19].

| | garden *kitchen* | bicycle *bonsai* | flowers *counter* | treehill *room* | stump |
|---|---|---|---|---|---|
| NerfStudio [53] | 26.21/*25.96*/+0.96% | 24.06/*23.61*/+1.90% | 21.43/*21.12*/+1.45% | 23.23/*22.85*/+1.66% | 26.09/*25.75*/+1.33% |
| | 29.87/*29.96*/-0.28% | 30.51/*30.52*/-0.02% | 26.91/*26.80*/+0.43% | 30.79/*30.56*/+0.76% | |
| Zip-NeRF [4] | 28.58/*28.18*/+1.40% | 26.35/*25.87*/+1.85% | 22.69/*22.34*/+1.56% | 24.43/*24.01*/+1.73% | 27.70/*27.32*/+1.37% |
| | 32.57/*32.39*/+0.55% | 34.89/*34.67*/+0.64% | 29.14/*28.90*/+0.82% | 33.22/*32.95*/+0.80% | |
| Gaussian Splatting [19] | 27.75/*27.37*/+1.40% | 25.62/*25.20*/+1.65% | 21.84/*21.60*/+1.14% | 22.75/*22.46*/+1.30% | 26.82/*26.48*/+1.29% |
| | 31.65/*31.36*/+0.92% | 32.29/*32.10*/+0.60% | 28.97/*28.97*/+0.03% | 31.38/*31.43*/-0.17% | |
| Mip-Splatting [4] | 27.85/*27.48*/+1.34% | 25.68/*25.30*/+1.51% | 21.93/*21.64*/+1.35% | 22.99/*22.64*/+1.52% | 26.88/*26.52*/+1.33% |
| | 31.55/*31.12*/+1.38% | 32.39/*32.18*/+0.65% | 29.07/*29.04*/+0.10% | 31.68/*31.55*/+0.42% | |

Table 1: **Mip-NeRF 360 [3] image downscale protocol**. We compare the manual downscaling with the default protocol using released downscaled images *(middle number)*. While NeRFs use released downscaled images, 3DGS-based methods [19] downscale images internally without compression. *Indoor scenes* and outdoor scenes have downscale factors of 2 and 4, respectively.

**Blender results.** From the results, the discrepancy is again small for most methods. However, for the Instant-NGP method [39], we can notice larger differences in PSNR, especially for 'drums', and 'ficus'. Note, that Instant-NGP [39] uses a black background for training and evaluation. We run additional experiments confirming this to be the case of the difference in the metrics. Since Tetra-NeRF [25] uses NerfStudio [53] and the codebase evolved since the time of the release, we can notice a slight drop in the performance.

## 4.2 Quality vs. computational cost

When comparing methods, image quality is important, but we also need to consider the computational cost of training. Some methods perform better but require more computational power, while others achieve good results with less computation. To compare and visualize this, we plot the performance of different methods against their computational resources in Figure 4. We measure computational cost as the training time multiplied by GPU memory use. The reasoning is that most methods can be sped up with more GPU memory or larger batch sizes, while low GPU memory usage allows training on cheaper GPUs. We also show the variance in performance and computational cost. For both the Mip-NeRF 360 [3] and Blender [37] datasets, performance variance is similar across methods. However, due to its adaptive nature, computational cost variance is much higher for methods based on Gaussian Splatting [19] compared to NeRF-based methods [37, 3, 4].

## 4.3 Mip-NeRF 360 evaluation protocol details

As stressed in the introduction, ensuring the same evaluation protocol is crucial in benchmarking. In this section, we demonstrate this by comparing two evaluation protocols that were used for the Mip-NeRF 360 dataset [3]. In Section 4.1, we have seen that NerfBaselines achieves consistently lower PSNR for Mip Splatting [67]. The reason is that while NeRFs train/evaluate on released downscaled

| PSNR↑ | barn | caterpillar | truck | lighthouse | playground | train | auditorium | ballroom | courtroom | museum | palace | temple |
|---|---|---|---|---|---|---|---|---|---|---|---|---|
| | Training Data | | | Intermediate | | | Advanced | | | | | |
| Instant NGP [39] | 25.90 | 21.72 | 22.85 | 21.65 | 23.33 | 20.01 | 20.67 | 21.62 | 19.44 | 15.19 | 19.09 | 17.84 |
| NerfStudio [53] | 26.40 | 21.71 | 23.37 | 20.85 | 24.69 | 20.43 | 20.77 | 22.68 | 20.24 | 17.84 | 17.68 | 17.06 |
| Zip-NeRF [4] | 29.26 | 23.94 | 25.09 | 23.07 | 27.13 | 22.19 | 24.52 | 25.45 | 22.17 | 19.34 | 19.11 | 20.58 |
| Gaussian Splatting [19] | 27.51 | 23.38 | 24.25 | 22.11 | 25.37 | 21.67 | 24.13 | 24.07 | 23.12 | 20.92 | 19.63 | 20.85 |
| Mip-Splatting [67] | 27.75 | 23.42 | 24.36 | 22.25 | 25.87 | 21.82 | 24.41 | 24.15 | 23.00 | 20.88 | 19.63 | 20.55 |
| Gaussian Opacity Fields [68] | 25.72 | 21.78 | 22.33 | 21.80 | 23.89 | 19.69 | 23.20 | 22.84 | 21.15 | 19.92 | 16.46 | 20.29 |
| **SSIM↑** | | | | | | | | | | | | |
| Instant NGP [39] | 0.772 | 0.633 | 0.770 | 0.765 | 0.696 | 0.657 | 0.761 | 0.652 | 0.640 | 0.471 | 0.668 | 0.689 |
| NerfStudio [53] | 0.794 | 0.666 | 0.797 | 0.768 | 0.755 | 0.693 | 0.771 | 0.705 | 0.673 | 0.648 | 0.640 | 0.678 |
| Zip-NeRF [4] | 0.884 | 0.802 | 0.864 | 0.849 | 0.880 | 0.814 | 0.877 | 0.835 | 0.790 | 0.746 | 0.718 | 0.805 |
| Gaussian Splatting [19] | 0.852 | 0.791 | 0.853 | 0.843 | 0.848 | 0.791 | 0.871 | 0.824 | 0.790 | 0.764 | 0.736 | 0.806 |
| Mip-Splatting [67] | 0.855 | 0.790 | 0.857 | 0.844 | 0.861 | 0.795 | 0.872 | 0.826 | 0.791 | 0.768 | 0.731 | 0.805 |
| Gaussian Opacity Fields [68] | 0.866 | 0.791 | 0.860 | 0.833 | 0.869 | 0.796 | 0.871 | 0.818 | 0.781 | 0.761 | 0.683 | 0.794 |
| **LPIPS↑** | | | | | | | | | | | | |
| Instant NGP [39] | 0.271 | 0.360 | 0.216 | 0.281 | 0.343 | 0.334 | 0.429 | 0.352 | 0.448 | 0.606 | 0.440 | 0.424 |
| NerfStudio [53] | 0.215 | 0.302 | 0.167 | 0.245 | 0.249 | 0.261 | 0.330 | 0.261 | 0.336 | 0.311 | 0.452 | 0.392 |
| Zip-NeRF [4] | 0.083 | 0.152 | 0.081 | 0.131 | 0.095 | 0.119 | 0.153 | 0.113 | 0.153 | 0.159 | 0.317 | 0.183 |
| Gaussian Splatting [19] | 0.160 | 0.190 | 0.108 | 0.156 | 0.170 | 0.171 | 0.193 | 0.101 | 0.165 | 0.160 | 0.350 | 0.222 |
| Mip-Splatting [67] | 0.161 | 0.197 | 0.109 | 0.159 | 0.155 | 0.172 | 0.196 | 0.098 | 0.165 | 0.158 | 0.354 | 0.226 |
| Gaussian Opacity Fields [68] | 0.140 | 0.187 | 0.099 | 0.181 | 0.142 | 0.164 | 0.194 | 0.107 | 0.168 | 0.152 | 0.443 | 0.234 |

Table 2: **Tanks & Temples [23] results.** We show the PSNR, SSIM [6], and LPIPS [70] (AlexNet) of various implemented methods. The first , second , and third values are highlighted.

images, the 3DGS [19] and Mip Splatting codebases [67] downscales from the large original images during training and evaluation (without storing them as JPEGs, thus avoiding compression artifacts). From the results presented in Table 1, we can see that while the difference is not large and the relative ordering is preserved, not using compression consistently gives results with larger PSNR. The difference is larger for outdoor scenes, where the downscale factor of 4 was used as opposed to indoor scenes with a downscale factor of 2.

## 4.4 Tanks & Temples evaluation

To demonstrate how NerfBaseline simplifies the transfer of existing methods to new datasets, we evaluate various integrated methods on the Tanks and Temples [23] dataset. For the dataset, we run COLMAP reconstruction [48, 49] with a simple radial camera model shared for all images. Afterward, we undistorted and downscaled images by a factor of 2. For NerfStudio [53], we run the Mip-NeRF 360 configuration (which from our experiments performs better on the dataset). The results are given in Table 2. As we can see the reconstructions are dominated by Zip-NeRF [4] for easier scenes, while for 'Advanced', there is no single best-performing method. We believe this is caused by NeRFs with its fixed capacity does not scale as well to larger scenes as 3DGS, where the capacity is adaptively increased.

## 4.5 Off-trajectory qualitative comparison

While test-set metrics enable an effective way of comparing different methods, they are insufficient to fully evaluate the perceived quality [29]. Rendering images from poses with varying distances from the training camera's trajectory provides a lot of insight into the robustness of the learned representations. Therefore, NerfBaselines provides a viewer and a renderer to enable visualising methods and rendering images/videos outside the train trajectory. In Figure 5, we compare various methods by rendering trained scenes both close to the training camera trajectory and far from it. Notice how in the second row Instant-NGP [39], and 3DGS methods [19, 67] cannot fill the sparsely observed sky, while NerfStudio [53] and Zip-NeRF [4] can achieve it thanks to space contraction. Also, notice how 3DGS methods [19, 67] are more blurred in less observed regions.

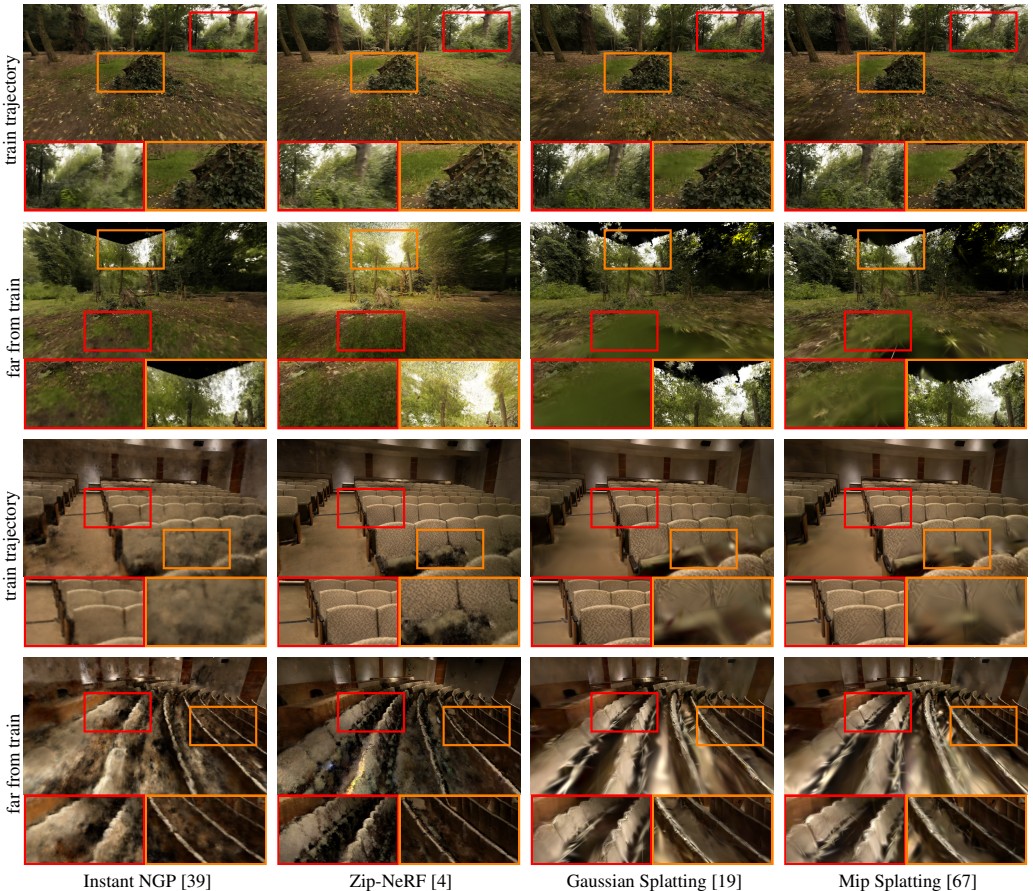

Figure 5: **Qualitative results.** We compare methods on views close and far from the training trajectory. **Top:** MipNeRF360/stump scene, **bottom:** T&T/Auditorium.

## 5 Conclusion

In conclusion, our NerfBaselines framework addresses the major challenges in evaluating novel view synthesis methods, e.g. NeRFs and 3DGS. By standardizing evaluation protocols and designing a unified interface, we enable fair comparisons and scalability to novel datasets. The camera trajectory editor enables multiview consistency evaluation, while the NerfBaselines framework ensures smooth installation and reproducibility by using isolated environments. Additionally, our web platform displays benchmark results, comparing various methods across different datasets. The NerfBaselines framework thus improves the fairness and effectiveness of novel view synthesis method evaluations.

**License.** This project is released under the **MIT** license. The integrated methods may be licensed under various licenses, and it is the user's responsibility to conform to the conditions before using a method. Note that while NerfBaselines enables access to various methods, the user still uses the original official codebases (with some wrapper code to interface with it).

**Limitations.** While NerfBaselines offers consistent and reproducible benchmarking, it requires methods to expose the same interface either directly or by writing a wrapper script. While we integrated some well-known methods and will gradually add more, our hope is that the scientific community could collaborate and adopt the interface for future methods.

**Broader impact.** While training and evaluating all methods require significant computational resources and associated carbon emissions, our release of reproducible checkpoints minimizes the need for repeated training by other researchers, thereby reducing overall environmental impact.

## Acknowledgments and Disclosure of Funding

We want to thank Brent Yi and the NerfStudio Team for helpful discussions regarding the NerfStudio codebase and for releasing the Viser platform. This work was supported by the Czech Science Foundation (GAČR) EXPRO (grant no. 23-07973X), the Grant Agency of the Czech Technical University in Prague (grant no. SGS24/095/OHK3/2T/13), and by the Ministry of Education, Youth and Sports of the Czech Republic through the e-INFRA CZ (ID:90254)

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
