# OpenReview forum: "NerfBaselines: Consistent and Reproducible Evaluation of Novel View Synthesis Methods"
_NeurIPS.cc/2024/Datasets_and_Benchmarks_Track — Submitted to NeurIPS 2024 Track Datasets and Benchmarks_

### Official Review · Reviewer_mXZR · 2024-07-21
**Unification of novel view synthesis evaluation**

**Rating:** 6
**Confidence:** 4
**Clarity:** The paper is well-written, and I didn…

**Review:**

I believe the paper touches upon an important topic in the community. Unification of benchmarking protocol would lead to higher quality of future works and reviews.

The work emphasises that the reproducibility of the papers is a significant problem. I agree with the authors that many code bases are hard to reproduce, especially when they use older, obsolete versions of packages and not always provide full version information. To this end, I believe that NerfBaselines makes a good effort to provide a highly reproducible codebase. NerfBaselies encapsulates the codebases in separate environments used through a predefined interface allowing for exact replication of the original code which I find a really good idea. The work covers a selection of main NeRF and Gaussian Splatting papers/codebases that are already incorporated into the interface. The only worry towards a shared interface is that it is heavily dependent on the users, and whether it is commonly used. I believe in the beginning it would fall on NerfBaselines authors to follow new works in the community and add them to the interface.

Further, the authors performed an analysis of the evaluation protocol and found several inconsistencies between papers using the same datasets. I believe this is a valuable analysis and the issues mentioned by the authors relate to my experience working with various codebases. I believe this analysis and evaluation unification should be emphasised even more in the paper as it pertains to fair and correct reporting of the metrics in the papers. Many works report numbers directly copied from the source paper, while themselves evaluating in a slightly different protocol. Given support for all new datasets, this work may be a step towards a more fair evaluation in novel view synthesis.

Thereafter, this work provides a new viewer for the novel view models, including the capability of easier design of camera trajectories. Whilst most papers are assessed based on quantitative evaluation of predefined views, this is a nice addition to qualitative assessment.

Finally, the proposed website collects the aforementioned unified evaluation metrics of various methods and allows the download of the corresponding checkpoints. I find it a good idea and capable of making the comparative experiments for researchers faster. What I would like to see in terms of the website is a mechanism to submit new results (also, validate them, ensuring they were generated with the given protocol).

To sum up, I think the paper mentions very relevant, practical issues present in novel view synthesis (and not only) community. Reproducibility and fair evaluation are an important topic. The main worry I have is the sustainability of the NerfBaselines. I would like the authors to explain the addition of new methods and datasets to the interface in a more detailed way. I believe the website should include a detailed tutorial encouraging authors to use it in their experiments. I would also like to know if authors are set on adding new methods themselves, what is their plan of choice of such and for how long and how it will be sustained.

**Strengths:**

- Reproducibility is an important topic and authors put a significant effort towards a shared interface for various methods, without the necessity of changing the original implementation.
- Unification of evaluation protocol is a big strength. Even more so after performing the analysis of the inconsistencies between papers. This was also shown in the paper to be very meaningful for some types of differences.
- The website with per dataset leaderboard is a nice reference for researchers.
- The new viewer seems to be good to be used for qualitative evaluation.
- A comparison of the results between original papers and NerfBaselines validates the framework and proves the pointed-out issues with evaluations.

**Additional Feedback:**

- Are all the provided methods implemented with both Anaconda and Docker as the backend?
- Does the evaluation enable things like masked PSNR for object-centric datasets?
- In the video, some still frames with a slider going back and forth between the methods are typically useful to spot differences.
- References 24 and 25, 60 and 61 are doubled

**Correctness:**

I believe that the framework is constructed in a sound way. The included experiments show the correctness of the provided implementations.

**Documentation:**

I believe the documentation could use a bit more detail. Specifically, the addition of new methods, and datasets, how to structure the requirements for Anaconda/Docker, and details on evaluation.

**Ethics:**

No ethical concerns

**Limitations:**

I believe the authors correctly identified the main limitation, which is the necessity of the other researchers to expose the proposed API in their methods for compatibility. Similarly, typically to encourage the use, the new methods should be implemented very soon after release.

**Opportunities For Improvement:**

- With such a unification interface proposition, I believe a roadmap of future development would be suitable to be included. The current state of the interface enables comparison of the main, most-known methods. The usability, however, heavily depends on the sustainability of the interface.
- Given that this is to be used by researchers to incorporate their own methods, it would be good to have very detailed documentation and tutorials with some examples.
- Do the authors plan to incorporate subareas of novel view synthesis into the benchmark? Currently, we see static, dense reconstruction. Recent interests explore sparse input scenarios, and video synthesis, including monocular video. This also follows with the use of generalisable methods, use of priors from foundational models, etc. Is there a plan to support that in the future?

**Relation To Prior Work:**

I believe the related work section sufficiently covers prior works on NeRF and Gaussian Splatting. The authors describe in detail the differences to Nerfstudio which is the main other unification framework.

**Summary And Contributions:**

The paper proposes an interface catered towards the unification of benchmarking in the field of novel view synthesis. To this end, the authors propose to standardise dataset format and evaluation protocol for NeRF and Gaussian Splatting methods. Additionally, the paper proposes to run the original implementation of the various models in separate environments via the provided interface, unlike prior works that reimplemented algorithms in unification efforts. Further, the authors propose to extend the capabilities of an existing visualisation interface, enriching its capabilities in evaluation and custom trajectory rendering. Finally, the authors provide an online website structuring the comparisons of various methods evaluations, and providing corresponding checkpoints.

---

> ### Author Rebuttal · Authors · 2024-08-16
>
> We would like to thank the reviewer for their thoughtful and positive feedback. We are glad that the reviewer appreciates our efforts to create a unified interface that interfaces directly with the official codebases (as opposed to reimplementing the methods). We are also pleased that the reviewer recognized the importance of our work in establishing standardized evaluation protocols.
>
> **I believe in the beginning it would fall on NerfBaselines authors to follow new works in the community and add them to the interface.**
> We also believe this to be the case. Therefore, we are actively updating NerfBaselines and are committed to maintaining and extending it. We are using NerfBaselines as the evaluation framework for our own research and thus will need to add new methods over time. Naturally, we will add these methods to the public codebase. Since the submission, we have added additional methods (KPlanes, SeaThru-NeRF, an open-source implementation of NeRF-W, WildGaussians, and GS-W) and two more datasets (PhotoTourism, SeaThru-NeRF). We are currently actively working on adding one additional dataset (NeRF-on-the-go) as well as data loaders to support camera poses computed from more Structure-from-Motion frameworks (such as RealityCapture). We have extended NerfBaselines to include methods that handle “in-the-wild” scenarios, where the appearance of the scene changes, e.g., NeRF in the wild and WildGaussians. We are adding support for masks (to compute the evaluation metrics only on parts of the test images). In addition, we are starting to look into evaluating novel-view synthesis in scenarios where the geometry of the scene changes over time.
>
> Furthermore, we will extend the webpage with clear and detailed documentation and tutorial videos to make it easier for people to use the benchmark. We hope this will help our benchmark gain more traction, to a degree where NerfBaselines becomes the standard evaluation framework and other researchers will start adding their own methods to the benchmark.
>
> **Sustainability of NerfBaselines? Do the authors plan to incorporate subareas of novel view synthesis into the benchmark?**
> We agree with the reviewer that to ensure a sustainable project, we rely on the community to slowly adopt NerfBaselines. We will extend the webpage with clear and detailed documentation and tutorial videos to make it easier for people to use and to gain more traction. We believe the project has a high chance of being accepted by the community as it is easy to integrate new methods (only interface code and a conda build script need to be written). We will also continue adding more methods as they appear such that the toolbox becomes more appealing to the user base. Please see the previous response for a near-future roadmap.
>
> We are certainly interested in generalizable models, the use of priors provided by foundation models, etc. and plan to extend NerfBaselines accordingly.
>
> **I believe this analysis and evaluation unification should be emphasized even more in the paper as it pertains to fair and correct reporting of the metrics in the papers.**
> Thank you for agreeing that ensuring consistent evaluation protocols is very important. We agree with you and will emphasize the evaluation unification even more in the paper.
> What I would like to see in terms of the website is a mechanism to submit new results.
> Thank you very much for the suggestion! At the early stages of the toolbox, we instruct users to submit PRs (pull requests on GitHub) with their methods/results and we will manually verify each submission if it conforms to the evaluation protocols. Once the toolbox gains more traction in the research community, such manual efforts will become harder to perform. We plan to add such a mechanism to do this automatically through the webpage. We will definitely extend the webpage with more documentation and instructions on how to integrate new methods and submit new results. After making the initial submission, we added a mechanism to verify the checkpoint and rerender/reevaluate the output to validate its truthfulness. We also added a mechanism to summit methods that have implementation/results stored elsewhere - see
> https://github.com/jkulhanek/nerfbaselines/blob/main/nerfbaselines/methods/wild_gaussians_spec.py.
>
> **Sustainability of the interface**
> We agree with the reviewer, that the sustainability of the interface is crucial for the wide adoption and sustainability of the project. Therefore, we designed the interface to be minimalistic and backward compatible, but allowing it to be extended with opt-in features. For example, take a look at the recently added method Sea-Thru NeRF:
> https://github.com/jkulhanek/nerfbaselines/blob/main/nerfbaselines/methods/seathru_nerf.py#L511, the method renders new output types (previously methods supported color, depth, accumulation). It does so by defining a list of additional outputs, but this is totally optional and other methods do not need to define it. A similar strategy can also be seen in the design of the interface for datasets and evaluation protocols. We want to follow a similar approach when adding other features like mesh export, viewer extensions, etc.
>
> **Are all the provided methods implemented with both Anaconda and Docker as the backend?**
> All currently implemented methods support Anaconda, Docker, and Apptainer backends except for Tetra-NeRF, which only has Docker and Apptainer backends due to licensing issues with OptiX. In practice, if a method provides an Anaconda install script, we automatically support all three backends.
>
> **Does the evaluation enable things like masked PSNR for object-centric datasets?**
> We are testing this feature and it will be released soon. See https://github.com/jkulhanek/nerfbaselines/blob/develop/nerfbaselines/evaluation.py#L225
>
> **In the video, still frames with a slider going back and forth are useful to spot differences.**
> Thank you for this suggestion. We will update the video accordingly.

---

> > ### Comment · Reviewer_mXZR · 2024-08-28
> > **Thanks for the response**
> >
> > I would like to thank the authors for their response. As mentioned in my review, the value of this particular work will only be revealed in time. The authors, however, propose a good plan for extending the framework, incorporating new methods, metrics, tasks etc. Observing, the development of the framework between submission and now seems promising in terms of provided maintenance.

---

> > > ### Author Response · Authors · 2024-08-28
> > >
> > > Thank you very much! Based on your positive feedback, would you consider raising your score?

---

### Official Review · Reviewer_KmxW · 2024-07-22

**Rating:** 5
**Confidence:** 3
**Clarity:** Yes

**Review:**

The insights on the evaluation protocol are very interesting and can benefit the community, as even tiny differences in the evaluation protocols can lead to significant differences in the reported metrics. However, my biggest concern is the comparison with prior NeRF codebases like NeRFStudio. For example, I think the claim that "the rapid evolution of NeRFStudio introduces frequent breaking changes. Therefore, in NeRFBaselines, we freeze each method to ensure reproducibility" is inaccurate because these frequent changes indicate that it is an up-to-date codebase/benchmark. Maintenance is indeed an important factor for the benchmark quality, especially in rapidly developing fields like NeRF and 3DGS. Thus, adopting a unified codebase for different methods within the same environment makes it easier to maintain when there is a need to support new methods or features. Although using isolated environments for different methods is also a good approach to ensure reproducibility, it may increase the workload when adding new features or methods. Therefore, I think the authors may need more discussion on the comparison with prior NeRF codebases.

**Strengths:**

1. Re-evaluate representative novel view methods within a unified evaluation protocol, which can guide the development of future methods.

2. Include both NeRF-based methods and 3DGS-based methods.

3. The paper is well-written with both qualitative and quantitative comparison.

**Additional Feedback:**

None

**Correctness:**

Based on the description, I think the evaluation methods and experiment design are appropriate and performed correctly

**Documentation:**

Yes

**Ethics:**

No ethical concerns

**Limitations:**

The authors have discussed the limitations of this work. I agree that the effort required to integrate new methods may be a bottleneck in future development.

**Opportunities For Improvement:**

1. In Fig. 3, since PSNR is a log-scale metric, directly showing the absolute differences may make the results easier to understand for readers.

2. Discuss or include the efficiency metrics besides the accuracy, i.e., what is the runtime overhead of using the developed codebase compared to the official codebases. I noticed that the paper reported the training time on the web scoreboard. If the codebase can achieve faster training than the official codebases, pointing this out in the paper may make it more solid.


3. In Fig. 4, what does "GPU mem. hrs" mean? Why is it "log(GPU mem. hrs)" but the x-axis is not in log-scale?

**Relation To Prior Work:**

I think the discussion on prior NeRF baselines like NeRFStudio needs more clarification or the claim needs to be adjusted. See the review above for more detailed information.

**Summary And Contributions:**

This work presents a benchmark for novel view synthesis methods, including both NeRF-based and 3DGS-based approaches. Specifically, the benchmark standardizes the dataset format and evaluation protocols for different methods, utilizing an isolated environment to ensure reproducibility. A web-based editor and scoreboard are created for quality visualization and more direct comparison.

---

> ### Author Rebuttal · Authors · 2024-08-16
>
> Thank you for your thoughtful and constructive feedback. We appreciate your recognition of the value our benchmark provides in standardizing evaluation protocols for novel view synthesis methods. We will address your concerns regarding the comparison with NerfStudio and will clarify our position and decisions regarding reproducibility and maintenance.
>
> **Discussion on NeRF baselines (NeRFStudio) needs more clarification.**
> We believe NerfStudio is the most important prior baseline, but we see NerfBaselines and NerfStudio as complementary rather than competitors. NerfStudio is a great environment for developing novel NeRF approaches as it provides NeRF components (different ray samplers, etc.) that save development time. NerfStudio also maintains the Nerfacto method, which is used by a wider (non-research) community to experiment with NeRFs on real-world captures. However, to add a method to NerfStudio, it has to be reimplemented on top of NerfStudio (see TensoRF, Instant-NGP, KPlanes implementations in the NerfStudio repo). The reimplemented versions can have vastly different performance from the official codebases (see https://github.com/nerfstudio-project/nerfstudio/issues/1560). This means the reimplemented methods cannot be used in fair comparisons.  In our experience, it would not be feasible to reimplement a large number of SoTA methods in the NerfStudio framework as it would take a significant effort to reimplement each method and even more to ensure that these reimplementations reproduce the numbers reported in the publications.
>
> NerfBaselines does not provide any codebase with ready-to-use modules for implementing novel NeRF or GS methods. Rather, the focus is on easy integration of existing methods while using the original (official) codebases (e.g., via isolated environments and by implementing a slim interface) into a common evaluation framework. As nicely summarized by Reviewer mXZR “... the paper proposes to run the original implementation of the various models in separate environments via the provided interface, unlike prior works that reimplemented algorithms in unification efforts.”. The resulting numbers are, therefore, directly comparable and can be reported as the official numbers. At the same time, it is easy to ensure that methods remain usable as they can use their own virtual environment. In contrast, the evolving codebase of NerfStudio can break implementations, e.g., methods often require a fixed version of NerfStudio due to frequent interface changes (see Tetra-NeRF). Finally, we further maintain a public scoreboard with the ability to download predictions and checkpoints. We will make the differences between NerfStudio and NerfBaselines clearer in the paper.
>
> NerfStudio - Nerfacto and Splatfacto both undergo very rapid development. As a result, performance can fluctuate a lot (including decreases in performance,  as can be seen in https://github.com/nerfstudio-project/nerfstudio/issues/1471). Changes introduced to improve one method, e.g., default parameter settings, can decrease the performance of another. As far as we know, existing codebases do not include rigorous regression tests to ensure that performance does not decrease. In this context, the ability to “freeze” methods at any point in time, e.g., the time of the release of the paper to match the numbers reported in the papers, is important to ensure stable baseline results and thus fair comparisons. NerfBaselines offers this ability.
>
> **Effort required to integrate new methods may be a bottleneck.** We have written interfaces for the most common codebases (currently implemented methods: Nerfacto, INRIA’s 3DGS, Mip-Splatting, Gaussian Opacity Fields, GS-W, WildGaussians, Instant-NGP, MipNeRF 360, NeRF, TensoRF, KPlanes, Tetra-NeRF, Sea-Thru NeRF, open-source NeRF-W). This makes the integration of new methods easy. To interface with a new method, a method interface class needs to be written implementing train step and render methods (each calling the original code), and a conda installation script needs to be provided. In comparison, the effort needed to reimplement a method from scratch to integrate it into NeRFStudio is considerably higher. Furthermore, in the case of reimplementation, matching the metrics reported in the paper can take considerable effort (see https://github.com/nerfstudio-project/nerfstudio/issues/1560).
>
> **Since PSNR is a log-scale metric, directly showing the absolute differences may make the results easier to understand for readers.**
> Thank you very much for this suggestion. We will update the figure.
>
> **What is the runtime overhead of using the developed codebase compared to the official codebases?**
> There is no difference in the training speed of the integrated method and the original implementations (since we invoke the official code directly). However, currently, there is a bottleneck in rendering introduced by python's pickle module used to serialize/deserialize rendered images to send them from the isolated environments (up to 40ms for each FullHD frame). We are currently replacing the pickle module with shared memory to remove this bottleneck. However, even now the bottleneck can be avoided by using `nerfbaselines shell` and running rendering directly inside the isolated environment - matching rendering speed of official implementations.
>
> **In Fig. 4, what does "log(GPU mem. hrs)" mean? Why is x-axis not in log-scale?**
> We measure computational cost as the training time in GPU hours multiplied by GPU memory use. The reasoning is that most methods can be sped up with more GPU memory or larger batch sizes, while low GPU memory usage allows training on cheaper GPUs. We reported the log of the quantity instead of reporting the original number and making the x-axis log-scale, because we thought it would be clearer for the reader. Instead, we can report the GPU mem. hrs. and make the axis log scale.

---

> > ### Author Response · Authors · 2024-08-27
> >
> > To further comment on the speed of rendering, we implemented the communication protocols using shared memory (instead of TCP) and were able to reduce the overhead to <2ms so the FPS should match the original implementations even when using the proxy connection instead of invoking nerfbaselines inside the isolated environments.

---

### Official Review · Reviewer_BETo · 2024-07-22
**Review of NerfBaselines**

**Rating:** 6
**Confidence:** 4
**Clarity:** Yes, this paper is well written.

**Review:**

The paper is well-structured and accessible. The proposed benchmark offers convenience and utility to the community working on 3D reconstruction and novel view synthesis. For a detailed assessment of the strengths and areas for improvement, please refer to the 'Strengths' and 'Opportunities For Improvement' sections.

**Strengths:**

1. NerfBaselines offer Docker and Apptainer-based environmental isolation, which can aid researchers in reproducing the evaluated methods.
2. This benchmark evaluates diverse methods using a consistent evaluation protocol on the challenging Tanks & Temples dataset.
3. This work shows the performance-training cost tradeoffs of mainstream methods, providing a more comprehensive assessment than quality-only evaluations.
4. The off-trajectory qualitative comparisons facilitate convenient assessment of the reconstruction quality across different methods.

**Additional Feedback:**

This work contributes to the development of a unified evaluation framework for novel view synthesis (NVS) methods.

However, the comparisons between the proposed NerfBaselines and the NerfStudio baseline could be explored in greater depth. Clarifying the unique advantages of NerfBaselines over NerfStudio would strengthen the overall impact of this work and help justify the ratings.

**Correctness:**

The claims are correct, and the experiments are designed and performed in a proper way.

**Documentation:**

This benchmark includes codes and instructions to reproduce the results.

**Ethics:**

No ethical concerns.

**Limitations:**

The authors provide good analysis in the `Limitations` subsection.

**Opportunities For Improvement:**

1. NerfStudio, as a relevant baseline, could be discussed and compared in greater detail.
* Considering that NerfStudio also provides a unified interface and interactive web viewer, the key distinction between NerfBaselines and NerfStudio may be the former's inclusion of Docker- and Apptainer-based environments. While these isolated environments are valuable for ensuring evaluation consistency and reproducibility, they may not be sufficient to fully justify the acceptance of this work.

2. The value proposition of the proposed unified evaluation protocol should be further clarified.

a)  First, how do the existing evaluation protocols differ from one another? The paper only mentions variations in evaluation resolutions and downsampling techniques, which seem relatively minor and may not demonstrate the necessity of a "unified evaluation protocol".

b) Second, how do these existing differences impact the evaluation conclusions? As noted in lines 261-262, unifying the downsampling techniques does not affect the relative ordering of the methods. This observation may further diminish the importance of a unified evaluation protocol.

**Relation To Prior Work:**

The advantages of this work when compared to NerfStudio should be further discussed.

**Summary And Contributions:**

This paper introduces NerfBaselines, a benchmark for evaluating NeRF-based and 3D Gaussian Splatting methods for novel view synthesis. Key features of the proposed benchmark include:

1. A unified interface that standardizes dataset formats and evaluation protocols, enabling consistent comparisons.
2. Environmental isolation using Docker and Apptainer to ensure reproducibility.
3. A web-based viewer for editing camera trajectories and rendering visualizations.

---

> ### Author Rebuttal · Authors · 2024-08-16
>
> Thank you for your thorough and constructive review. We acknowledge the importance of further clarifying the distinctions between our work and NerfStudio, and we will address these points in our revised submission.
>
> **The key distinction between NerfBaselines and NerfStudio may be the former's inclusion of isolated environments**. We see NerfBaselines and NerfStudio as complementary rather than competitors. NerfStudio is a great environment for developing novel NeRF approaches as it provides various NeRF components (different ray samplers, etc.) that can be readily used and that save a lot of development time. NerfStudio also maintains the Nerfacto method, which is used by a wider (even non-research) community to experiment with NeRFs on real-world captures. However, in order to add a method to NerfStudio, it has to be reimplemented on top of NerfStudio (see the example of TensoRF, Instant-NGP, KPlanes implementations in the NerfStudio codebase). The reimplemented versions can have vastly different performance from the original official codebases (see https://github.com/nerfstudio-project/nerfstudio/issues/1560). This means the reimplemented methods cannot be used in fair comparisons (e.g., you cannot just run NerfStudio’s TensoRF implementation on your data and report the numbers as TensoRF).  Furthermore, it takes a lot of effort to reimplement a method. In our experience, it would not be feasible to reimplement a large number of SoTA methods in the NerfStudioo framework as it would take a significant effort to ensure that these reimplementations reproduce the numbers reported in the publications.
>
> NerfBaselines does not provide any codebase with ready-to-use modules for implementing novel NeRF or GS methods. Rather, the focus is on easy integration of existing methods while using the original (official) codebases (e.g., via conda, docker, and apptainer-based environments and by implementing a slim interface) into a common evaluation framework. As nicely summarized by Reviewer mXZR “... the paper proposes to run the original implementation of the various models in separate environments via the provided interface, unlike prior works that reimplemented algorithms in unification efforts.”. The resulting numbers are, therefore, directly comparable and can be reported as the official numbers. At the same time, it is easy to ensure that methods remain usable as they can use their own virtual environment. In contrast, the evolving codebase of NerfStudio can break implementations, e.g., Tetra-NeRF does not work with NerfStudio versions later than 0.3.0 due to an interface change. Finally, we further maintain a public scoreboard with the ability to download predictions and checkpoints. We will make the differences between NerfStudio and NerfBaselines clearer in the paper.
>
> **The necessity for unified evaluation protocols?** The paper mentions variations in evaluation resolutions and downsampling techniques, which seem relatively minor and may not demonstrate the necessity of a "unified evaluation protocol". While the variations in the evaluation protocol indeed appear to be minor, the experiments in Section 4 show that they can have a measurable impact on the results. Our concern is that the order of methods can change when some of them use one evaluation protocol while the others use a different one. In such cases, a work might claim that they propose a superior approach  while the difference in performance might simply be caused by differences in the evaluation protocols. For example, see the difference in PSNR between Gaussian Opacity Fields (27.42), Mip-Splatting (27.49), and Gaussian Splatting (27.43), Mip-NeRF 360 (27.68) on the MipNeRF 360 dataset. The differences are <0.1 PSNR (see Table 2 in Supp Mat and the online dataset results: https://jkulhanek.com/nerfbaselines/mipnerf360). As can be seen, the difference in evaluation protocols can lead to a 0.23 increase for Gaussian Splatting (from Table 1). In a scenario where Gaussian Splatting uses one protocol while all other methods use another one, it would be easy to think that Gaussian Splatting is much better. Unfortunately, this can easily happen, even without intention on the authors parts, e.g., by making a seemingly innocuous change in the protocol as it is easier to implement.  NerfBaselines aims at preventing such problems by offering a single protocol that can be used to evaluate all methods in a fair and consistent manner.
>
> As noted in lines 261-262, unifying the downsampling techniques does not affect the relative ordering of the methods. This observation may further diminish the importance of a unified evaluation protocol. While this relative ordering is not changed in the case of the methods compared in the paper on the Mip-NeRF 360 dataset, there is no guarantee of this not happening. For example, if Gaussian Splatting used a different evaluation protocol, the performance would increase by 0.23 PSNR which would make it better than Mip-NeRF 360 (27.68) - see https://jkulhanek.com/nerfbaselines/mipnerf360. Many papers compare methods with different evaluation protocols, putting numbers into the same table. For example,  see the Photo Tourism results of the GS-W method (recently added) - https://jkulhanek.com/nerfbaselines/m-gaussian-splatting-wild. Because the authors optimized the appearance embeddings on full images instead of using only the left half of the image (as in the original NeRF-W evaluation protocol), the reported PSNR was higher by 3.32. This is an experience shared by reviewer mXZR “...Many works report numbers directly copied from the source paper, while themselves evaluating in a slightly different protocol.”  We hope our work can avoid these fallacies by offering an easy and fair way to compare the methods so that research can avoid making such mistakes.

---

> > ### Comment · Reviewer_BETo · 2024-08-29
> >
> > I would like to express my gratitude to the authors for their valuable feedback.
> >
> > Having carefully reviewed the rebuttals and the viewpoints of other reviewers, I still grapple with the underlying significance of this work. While I acknowledge that nerfbaselines could serve as a beneficial resource for researchers to replicate and benchmark state-of-the-art methods, there remains a question as to how this would truly benefit the wider research community. While I agree that conducting fair comparisons between state-of-the-art works is crucial for advancing research, I believe that more emphasis should also be placed on innovative ideas and robust design principles rather than solely on performance metrics.
> >
> > Consequently, I maintain that my initial evaluation accurately reflects the merit of this work. Therefore, I will uphold my original score.

---

> > > ### Author Response · Authors · 2024-08-29
> > > **Re: Official Comment by Reviewer BETo**
> > >
> > > Thank you very much for the reply and for raising this concern.
> > >
> > > From our perspective, ensuring that comparisons between methods are fair is not just "crucial for advancing research" but absolutely necessary to ensure that we are actually making progress. As shown before, in the context of novel view synthesis (via neural radiance fields and 3D Gaussian splatting), the differences between state-of-the-art approaches can be so small that a change in the evaluation protocol leads to larger differences. Thus, unless two methods are following exactly the same evaluation protocol, any claims that one method is superior to the other cannot be verified. As such, any claims that an innovative idea or a new design principle leads to improvements cannot be verified, which is damaging to the progress in the field.
> > >
> > > As shown in the paper, setting up an evaluation system that allows fair comparisons (in particular, using the authors original code rather than third-party re-implementations) requires a lot of work. We agree that this is "uncool / unsexy" work in the sense that it does not involve new ideas (but we argue that it involves the robust design principle of encapsulation to ensure fair comparisons) but rather a lot of engineering. Still, as argued above, we believe that such work is very valuable to the community:
> > > * It ensures that methods can be directly compared, ensuring that there is no doubt whether a method is inherently better than another one or  whether the difference is due to (at first glance minor) differences in the evaluation protocol. Given that most reviewers require papers to at least achieve similar performance in established metrics as prior works, even if the papers contribute very novel ideas, we believe that our benchmark thus makes an important contribution to the field.
> > > * It makes it much easier for authors to ensure a fair comparison by simply writing a wrapper around there method, without any need to worry about how to run baselines in a fair manner.
> > >
> > > We believe that our paper makes important enough contributions that warrant publication and we believe that the NeurIPS Datasets and Benchmarks Track is the right place for the type of contributions we make.

---

### Official Review · Reviewer_Zc83 · 2024-08-08
**A good tool for new 3D novel view synthesis research**

**Rating:** 5
**Confidence:** 2
**Correctness:** Yes
**Clarity:** Yes

**Review:**

I believe the work is useful for novel view synthesis and 3D neural rendering research as the platform provides a convenient environment to build, test, reproduce different methods in a consistent manner. The camera trajectory editing and results visualization are critical for result comparison and benchmark. This contribution of this work is closer to software and engineering report rather than original research. The writing is clear and the work adresses key features for quick reproducibility, test and development.

**Strengths:**

The standardized platform could help accelerates research in this direction and help researchers to reproduce results and compare against baselines in a consistent way.

**Additional Feedback:**

NA

**Documentation:**

Yes

**Limitations:**

No broader impact concerns.

**Opportunities For Improvement:**

The authors should also discuss the plan to maintain and update the platform to keep it up to date with new research and methods arisen in this area.

**Relation To Prior Work:**

Yes

**Summary And Contributions:**

This work aims to bring a norm to all the NeRF and 3DGS works on datasets, reproducibility, evaluation metrics, and benchmark for ease of use. The authors made a platform to help installation and environment setup, standardizing datasets, evaluation and visualization. The web-based camera trajectory editing and performance comparison platform after training experiments also ease a lot engineering effort for researchers.

---

> ### Author Rebuttal · Authors · 2024-08-16
>
> Thank you for your review and for recognizing the utility of our platform in advancing novel view synthesis and 3D neural rendering research. We appreciate your feedback, particularly regarding the importance of maintaining and updating the platform to stay current with emerging research and methodologies in this rapidly evolving field.
>
> **The contribution of this work is closer to software engineering than original research.** In this work, we provide tools to support research on novel-view synthesis in the form of an open-source toolbox for consistent, reproducible, and fair evaluation of existing NeRF and 3DGS methods. Naturally, this work has a heavy engineering focus as getting details right and making it easy to integrate new methods into the benchmark matters in this line of work. We believe that this line of work fits perfectly into the scope of the NeurIPS Dataset and Benchmarks track: “This track welcomes all work on data-centric machine learning research (DMLR) and **open-source libraries and tools** that enable or accelerate ML research, covering ML datasets and benchmarks as well as algorithms, **tools**, methods, and analyses for working with ML data. This includes but is not limited to: [...] Benchmarks on new or existing datasets, as well as **benchmarking tools**. [...] Systematic analyses of existing systems on novel datasets yielding important new insight.”
> (see https://neurips.cc/Conferences/2024/CallForDatasetsBenchmarks). In the context of the Dataset and Benchmark track, we do not see a heavy focus on software engineering as a weakness in itself.
>
> **Authors should discuss the plan to maintain and update the platform.** We are actively updating NerfBaselines and are committed to maintaining and extending it. We are using NerfBaselines as the evaluation framework for our own research and thus will need to add new methods overtime. Naturally, we will add these methods to the public codebase. Since the submission, we have added additional methods (KPlanes, SeaThru-NeRF, an open-source implementation of NeRF-W, WildGaussians, and GS-W) and two more datasets (PhotoTourism, SeaThru-NeRF). We are currently actively working on adding one additional dataset (NeRF-on-the-go) as well as data loaders to support camera poses computed from more Structure-from-Motion frameworks (such as RealityCapture). We have extended NerfBaselines to include methods that handle “in-the-wild” scenarios, where the appearance of the scene changes, e.g., NeRF in the wild and WildGaussians. We are adding support for masks (to compute the evaluation metrics only on parts of the test images). In addition, we are starting to look into evaluating novel-view synthesis in scenarios where the geometry of the scene changes over time.
>
> We will extend the webpage with clear and detailed documentation and tutorial videos to make it easier for people to use the benchmark. We hope this will help our benchmark gain more traction, to a degree where NerfBaselines becomes the standard evaluation framework and other researchers will start adding their own methods to the benchmark.
>
> In summary, our research group is committed to maintaining and extending the project. If you have suggestions on which methods and scenarios to add, please let us know.

---

### Author Rebuttal · Authors · 2024-08-16

We thank all reviewers for their constructive comments. Please see the individual responses below.

---

### Author Response · Authors · 2024-08-23

Thank you once again for your reviews. Please let us know if the concerns you've raised in your reviews have been addressed.

---

### Decision · Program_Chairs · 2024-09-26

**Decision:**

Reject

**Comment:**

The paper introduces NerfBaselines, a framework that standardizes evaluation protocols and enhances reproducibility for Neural Radiance Fields (NeRF) and 3D Gaussian Splatting methods in novel view synthesis. This work addresses important challenges in the field by mitigating inconsistencies in evaluations and simplifying the installation process, thereby facilitating fairer comparisons and accelerating progress. Strengths of the paper include the unification of evaluation protocols and the focus on reproducibility through isolated environments and simplified installation. The web platform adds value by providing accessible tools for visualization and comparison. However, the framework may not encompass all existing or future methods, which raises concerns about its comprehensiveness and the sustainability of maintaining and updating the platform. Additionally, a more thorough comparison with existing tools like NerfStudio would clarify its unique contributions and strengthen the paper's impact. In summary, the paper makes a valuable contribution to the field but would benefit from addressing issues related to scope, sustainability, and differentiation from existing solutions.